

# Electroantennogram and Olfactory behavioral responses of *Trabala vishnou gigantina* (Lepidoptera: Lasiocampidae) to herbivore induced *Hippophae rhamnoides* leaf volatiles

Yonghua Liu, Kexu An, Shuo Tang, Jiangshuai Feng and Xiongfei Yan

Key Laboratory of Ecological Restoration in Shanbei Mining Area, and Key Laboratory of Plant Pest Control in Yulin City, Yulin University, Yulin, China

## ABSTRACT

The moth *Trabala vishnou gigantina* Yang, 1978 (Lepidoptera: Lasiocampidae), a leaf-eating pest, had a severe outbreak in the sea buckthorn, *Hippophae rhamnoides* L. (Elaeagnaceae) plantation in North China. This study aims to investigate the influence of volatiles emitted by *T. vishnou gigantina*-infested *H. rhamnoides* on the behavioral responses of *T. vishnou gigantina*, laying a basis for the development of plant-based elicitors. The chemical basis of herbivore insect host dynamics between these species were investigated, providing information for herbivore insect control methods. After identifying the compounds produced by the differences between healthy and infested plants, six critical volatile compounds were identified to explore their attractions to *T. vishnou gigantina* imagoes by using electroantennogram (EAG) and olfactory behavioral response experiments. The results showed that the EAG responses of *T. vishnou gigantina* imagoes did not only vary in these six different herbivore-infested volatiles, but also between different concentrations of the same volatile. The EAG responses to the herbivore-infested volatiles peaked at a stimulus concentration of 100 µg/µL, with Hexyl 2-methyl butyrate having the highest EAG response. The test results of olfactory behavioral responses had significant differences in the olfactory behavioral responses of female and male imagoes to the six herbivore-infested volatiles. Hexyl 2-methyl butyrate had the strongest attraction to female and male imagoes, followed by 2-ethylhexanol and longifolene. 2-ethylhexanol had a significant attraction to female imagoes, while there was no significant attraction to males.

## INTRODUCTION

*Hippophae rhamnoides* is a member of the Elaeagnaceae family, distributed in northwest, southwest, and north China. It is highly resistant to cold, drought, wind, and sandy conditions. In northwest China, it is widely used for desert greening. Its root system develops together with nitrogen-fixing bacteria, reducing wind speed and erosion, and having a positive effect on soil ecosystems (*Yang et al., 2024*). It has extremely high

Corresponding author
Yonghua Liu, liuyong2130@yeah.net

practical value in soil and water conservation. Larvae of the moth *T. vishnou gigantina* Yang (Lepidoptera: Lasiocampidae) feed on *H. rhamnoides* leaf blades and leave only petioles, resulting in the failure of normal physiological functions of plants (*Liu et al., 2021a*). *H. rhamnoides* is not only used for food production, but also for soil improvement. In recent years, *T. vishnou gigantina* outbreaks have occurred in the sea buckthorn planting areas in Wuqi County and Zhidan County in Shaanxi Province, China, severely reducing plant vigor and threatening the healthy and sustainable development of local *H. rhamnoides* industry (*Liu et al., 2013*).

Host volatiles are a major and important regulator in the plant-herbivore nutrient system (*Zang et al., 2021*). Lure or repel insects of the same or different species, herbivore infestation can induce host plants to generate new volatiles or cause changes in the composition of host volatiles, thereby affecting the growth, reproduction, migration and population of herbivore species (*Qian et al., 2024*; *Liu et al., 2021b*). Elucidating the interactions between herbivores and herbivore-infested host volatiles can provide an important basis for developing sustainable solutions to herbivore insect control (*Arriola et al., 2020*; *Mitra et al., 2020*). For example, *Qiao, Cai & Su (2020)* found that aphid-induced wheat volatiles can attract *Harmonia axyridis*, a natural enemy of aphids, making them an effective solution for herbivore insect control in agriculture. Therefore, studying the interactions between herbivores and host volatiles can facilitate the identification of compounds with attractant or repellent activity (*Turlings & Erb, 2018*).

To date, several studies have investigated the host selection mechanism of *T. vishnou gigantina*, its bioecological characteristics (*Liu et al., 2013*), electrophysiological and behavioral responses to volatiles of *H. rhamnoides* (*Liu et al., 2022*), as well as the prevention and control methods (*Liu et al., 2021a*). However, few studies have examined the electroantennogram and olfactory behavioral responses of *T. vishnou gigantina* to herbivore-infested *H. rhamnoides* volatiles (*Niu, Xu & Lin, 2024*). Therefore, we analyzed the composition of *T. vishnou gigantina*-induced *H. rhamnoides* volatiles and identified a total of 34 compounds. The composition of *H. rhamnoides* volatiles in healthy and infected individuals showed significant changes. Six compounds that may significantly affect *T. vishnou gigantina* were identified. Experiments were conducted on electroantennogram (EAG) and olfactory behavioral response, providing a theoretical basis for the prevention and control of *T. vishnou gigantina* using plant-derived attractants or repellents.

## MATERIALS AND RESEARCH METHOD

### Test insects

In mid-August 2023, about 200 cocoons of *T. vishnou gigantina* were collected from *H. rhamnoides* forests in Wuqi County, Shaanxi Province, China to place them in an insect cage ($100 \times 40 \times 50$ cm). After hatching the cocoons, unmated male and female imagoes were distinguished, 3 days old and healthy and equally sized imagoes were selected for experiment.

## Collection and analysis of *H. rhamnoides* volatiles
### Collection of H. rhamnoides samples

Fresh, healthy and infested *H. rhamnoides* leaf blades were collected from the experimental site. Healthy leaves were obtained from plants that were not damaged by *T. vishnou gigantina*, placed in centrifuge tubes. They were rapidly frozen with liquid nitrogen, labeled, and then back to the laboratory for test. Each group were repeated three times. The time for collecting samples was the same as that for behavioral test using a Y-tube olfactometer.

### Extraction and analysis of H. rhamnoides volatiles

*H. rhamnoides* leaf samples were stored in a refrigerator at −80 °C and ground with liquid nitrogen and mixed evenly during the operational test. A 500 mg subsample was taken from each sample and placed it into a headspace vial, and then a saturated NaCl solution was added to prepare a 20 µL internal standard solution. Fully automated headspace solid phase microextraction HS-SPME (CTC Analytics AG) was used to extract samples for GC-MS analysis. Then, the internal standard solution was shaked at 60 °C for 5 min, and a 120 µm Agilent SPME Fiber head (DVB/C-WR/PDMS) extraction head was inserted into the sample headspace of a Fiber Conditioning Station (CTC Analytics AG) before sampling, and shaked at 250 °C for 5 min. The headspace was extracted for 15 min and resolved at 250 °C for 5 min, and then separated and determined by a GC-MS. The chromatographic column was a DB-5MS capillary column (30 m × 0.25 mm × 0.25 µm; Agilent J&W Scientific, Folsom, CA, USA), and the carrier gas was high-purity helium. Splitless sampling was conducted at a flow rate of 1.2 mL/min. The sample inlet port was heated to 250 °C, and the solvent was delayed for 3.5 min. The temperature was programmed as 40 °C for 3.5 min, 10 °C/min to 100 °C, 7 °C/min to 180 °C, 25 °C/min to 280 °C, and hold at 280 °C for 5 min. The MS test conditions were: EI ion source, ion source temperature of 230 °C, electron energy of 70 eV, quadruple rod temperature of 150 °C, MS interface temperature of 280 °C, and ion detection mode (SIM). The total ion chromatogram (total ion current, TIC) of *H. rhamnoides volatiles* was used to obtain MS data using Mass Hunter software for qualitative and quantitative analysis (*Nusra et al., 2021*).

## Test volatiles and preparation

Six key volatiles were screened by volatile data analysis. Table 1 shows names, purity and sources of these six standard compounds. These compounds were dissolved in liquid paraffin, and each was prepared in 0.1, 1, 10, 50, and 100-µg/µL solutions.

## EAG response (mv) test

EAG is a widely used biological identification instrument in experimental entomology. It is mainly used to detect insects that perceive the world through their antennae. Healthy and highly active *T. vishnou gigantina* imagoes aged 1–3 day were selected, and the test apparatus were sterilized before use. An antenna was removed from the base of each imago using a scalpel, and one flagellum was removed from the end of the antenna. The two ends of the antenna were in full contact with two electrodes using conductive adhesive,

**Table 1** The names, purity, and sources of six volatile substances tested.

| Standard compounds | Purity | Source of supply |
|---|---|---|
| Hexyl 2-methyl butyrate | 99% | Aladdin |
| 2-ethylhexanol | 99% | Aladdin |
| Humulene | 98% | Sigma |
| Leaf acetate | 99% | Sigma |
| Aristolochene | 98% | Sigma |
| Longifolene | 98% | Sigma |

and the output end was connected to a host computer (Syntech UN-06).The setup was covered with a transparent hood to maintain stable experimental conditions. The sample compounds prepared were tested separately. The flow rate of the test gas was 400 mL/min, and the measurement was recorded after baseline stabilization. The stimulation time was set to 0.5 s, with an interval of at least 30 s to ensure that the sensory function of the antennal sensilla was completely restored. Different compounds were tested at the same concentration on six male and female imagoes separately from each other. Experimental insects cannot be reused, and male and female adults were tested separately. Each sample was stimulated with an average of five times, and repeated three times. During testing, the concentration of the standardized compounds increased from low to high. When the same concentration gradient of the tested compounds was completed, the clean test tubes were replaced. Liquid paraffin was used as the blank control. The data was collected and analyzed using Spike software (Syntech). The EAG measurements were read out directly by an antenna potentiometer. The relative EAG response was calculated as follows:

Relative EAG response (mv) = (Relative value of EAG response of test sample) (mv) -[(pre-test EAG response of control group) (mv) + (post-test EAG response of control group) (mv)]/2.

## Olfactory behavioral response bioassay by Y-tube olfactometry

The behavioral response of *T. vishnou gigantina* to the six herbivore-infested *H. rhamnoides* volatiles was tested using a Y-tube olfactometer with an inner diameter of 3 cm, a main arm of 15 cm, two side arms of 8 cm (with an angle of 75° between them, and an air flow rate of 400 mL/min. To ensure uniform illumination of all parts, a light source was placed 30 cm above the olfactometer. The highest concentration in the EAG response test (100 µg/µL) was selected, and 20 µL of each test sample was dropped onto a strip of filter paper using a pipette and placed at the end of one side arm. The same volume of liquid paraffin was dripped onto another strip of filter paper and placed at end of the other arm as the control. If the *T. vishnou gigantina* entered the position at more than two-thirds of the attractor arm or control arm and stayed for more than 1 min, it was recorded as having a smell source tendency; otherwise, it was recorded as having no response. The olfactory behavioral responses of *T. vishnou gigantina* to volatiles of the same concentration were recorded as a group. After each test group, the test apparatus was cleaned and dried. The effects of the experimental environment were eliminated by interchanging the attractor arm with the control arm. 35–40 imagoes were tested under each volatile concentration.

## Data analysis and statistics

The relative content (percentage) of each volatile was calculated using the area normalization method. The generated data were statistically analyzed using the IBM SPSS Statistics 26. The significance of EAG differences between different concentrations of the same standard compound was calculated using the Duncan multiple comparisons method. The results of olfactory behavioral responses were analyzed using the $\chi^2$ test.

# RESULTS

## Collection and identification of volatiles from herbivore-infested *H. rhamnoides* leaves

33 volatiles were detected in the herbivore-infested *H. rhamnoides*, including six alcohols, three hydrocarbons, eleven terpenes, two ketones, four aldehydes, and seven esters (Table 2). Three new volatile compounds have been added and compared to healthy *H. rhamnoides* leaves, namel, humulene (2.42 ± 0.76) %, aristolochene (1.92 ± 0.42) %, and hexyl 2-methyl butyrate (0.58 ± 0.09) %. In addition, the relative content of three volatile compounds, leaf acetate (35.09 ± 1.12) %, 2-ethylhexanol (22.91 ± 3.09) %, and longifolene (2.41 ± 0.65) % significantly increased. Therefore, these six volatile compounds were selected for electroantennogram and olfactory behavioral response tests (*Sun, Gao & Chen, 2012*).

## EAG response of *T. vishnou gigantina* to herbivore-infested *H. rhamnoides* volatiles at different concentrations

EAG responses of *T. vishnou gigantina* imagoes varied not only in these six different herbivore-infested volatiles, but also between different concentrations of the same volatile (Table 3). At a concentration of 0.1 µg/µL, the EAG responses to all volatiles were weak. When the concentration increased to 1 µg/µL, compared with the concentration of 0.1 µg/µL, the relative EAG response of female and male *T. vishnou gigantina* imagoes increased significantly ($P < 0.05$). When the volatile concentration increased to 10 µg/µL, the relative EAG of hexyl 2-methyl butyrate and longifolene of *T. vishnou gigantina* females were significantly increased ($P < 0.05$). At a concentration of 50 µg/µL, the relative EAG responses of imagoes were all significantly higher than those at a concentration of 10 µg/µL ($P < 0.05$), with hexyl 2-methyl butyrate having the highest relative values. When the volatile concentration increased to 100 µg/µL, the relative values of all EAG responses were maximized, except for those of hexyl 2-methyl butyrate and Leaf acetate, which showed no significant difference from the values at a concentration of 50 µg/µL ($P > 0.05$).

Therefore, at a concentration of 100 µg/uL, the relative EAG response of *T. vishnou gigantina* adults was the highest. The relative EAG response of *T. vishnou gigantina* to different volatiles were also compared. The responses to hexyl 2-methyl butyrate were the highest (2.53 and 2.24), followed by those to longifolene (2.44 and 2.18), and those to Leaf acetate were the lowest (1.59 and 0.78). Overall, the relative EAG response of female and male *T. vishnou gigantina* imagoes to the six herbivore-infested *H. rhamnoides* volatiles continued to increase with increasing concentrations, with females consistently showed stronger responses than males.

**Table 2  Relative content of volatiles from healthy and herbivore-infested of *H. rhamnoides*.**

| Volatile types | Plant volatile | CAS | Relative content% | |
| --- | --- | --- | --- | --- |
| | | | Healthy *H. rhamnoides* | Herbivore-infested *H. rhamnoides* |
| Alcohol | 2-Heptanol | 6033-23-4 | 1.21 ± 0.17 | 1.23 ± 0.17 |
| | cis-5-Octen-1-ol | 64275-73-6 | 1.79 ± 0.73 | 1.78 ± 0.53 |
| | 1-Octanol | 111-87-5 | 10.41 ± 2.75 | 10.51 ± 2.01 |
| | 1-Octen-3-ol | 3391-86-4 | 1.47 ± 0.04 | 1.46 ± 0.32 |
| | Benzeneethanol, beta -methyl- | 1123-85-9 | 15.07 ± 0.37 | 15.71 ± 0.77 |
| | 2-ethylhexanol | 104-76-7 | 2.03 ± 0.05 | 22.91 ± 3.09[**] |
| Aldehydes | 1H-Pyrrole-2-carboxaldehyde | 1003-29-8 | 2.39 ± 0.27 | 2.25 ± 0.21 |
| | 1-Cyclohexene-1-acetaldehyde, 2,6,6-trimethyl- | 472-66-2 | 5.68 ± 1.02 | 5.96 ± 1.26 |
| | 10-Undecenal | 112-45-8 | 10.97 ± 2.66 | 10.94 ± 4.28 |
| | Octanal | 124-13-0 | 1.97 ± 0.23 | 1.84 ± 0.54 |
| Terpenoids | Aristolochene | 26620-71-3 | – | 1.92 ± 0.42 |
| | 10,10-Dimethyl-2,6-dimethylenebicyclo [7.2.0] undecane | 357414-37-0 | 6.56 ± 1.22 | 6.81 ± 1.89 |
| | Sabinyl acetate | 3536-54-7 | 2.79 ± 0.26 | 2.63 ± 0.33 |
| | (-)-$\alpha$-cedrene | 469-61-4 | 2.25 ± 0.12 | 2.36 ± 0.18 |
| | $\beta$-sesquiphellandrene | 20307-83-9 | 2.53 ± 0.43 | 2.47 ± 0.44 |
| | $\gamma$-cadinene | 39029-41-9 | 4.79 ± 1.27 | 4.93 ± 1.19 |
| | Linalool | 78-70-6 | 1.62 ± 0.21 | 1.71 ± 0.47 |
| | Humulene | 6753-98-6 | – | 2.42 ± 0.76 |
| | Cedrene | 11028-42-5 | 4.11 ± 1.13 | 4.14 ± 1.58 |
| | Longifolene | 475-20-7 | 0.23 ± 0.08 | 2.41 ± 0.65[*] |
| | Perillyl alcohol | 536-59-4 | 1.57 ± 0.46 | 1.62 ± 0.64 |
| Hydrocarbons | 1-Tridecene | 2437-56-1 | 4.70 ± 1.21 | 4.58 ± 1.48 |
| | Nonane | 111-84-2 | 1.36 ± 0.37 | 1.59 ± 0.41 |
| | p-cymene | 99-87-6 | 3.41 ± 1.05 | 3.54 ± 1.17 |
| Ketone | 2-Hexanone,3-methyl- | 2550-21-2 | 12.18 ± 3.31 | 12.41 ± 3.13 |
| | 5-Nonanone | 502-56-7 | 5.64 ± 1.31 | 5.47 ± 1.34 |
| Ester | Acetic acid,non-3-enyl ester, cis- | 13049-88-2 | 2.58 ± 0.29 | 2.35 ± 0.31 |
| | Hexyl 2-methyl butyrate | 10032-15-2 | – | 0.58 ± 0.09 |
| | Dihydroactinidiolide | 15356-74-8 | 11.68 ± 3.01 | 11.52 ± 3.72 |
| | Benzeneaceticacid, ethyl ester | 101-97-3 | 4.22 ± 1.34 | 4.28 ± 1.61 |
| | Butanoicacid, butyl ester | 109-21-7 | 17.51 ± 6.21 | 17.80 ± 6.33 |
| | Leaf acetate | 3681-71-8 | 3.09 ± 0.42 | 35.09 ± 1.12[**] |
| | cis-3-Hexenyl pyruvate | 68133-76-6 | 17.14 ± 7.26 | 17.42 ± 7.24 |

**Notes.**
The data in the table are mean ± SE, the dash (–) indicates that the compound was undetected, * and ** indicate significant differences in the relative contents of herbivore t-infested *H. rhamnoides* leaves compared with in healthy leaves at the 0.05 and 0.01 levels, respectively.
CAS, Chemical Abstracts Service Number.

**Table 3** Relative EAG response values of *T. vishnou gigantina* to different concentrations of volatile compounds from herbivore-infested *H. rhamnoides*.

| Compounds | Sex | Relative EAG response (μ g/μ L) | | | | |
|---|---|---|---|---|---|---|
| | | 0.1 | 1 | 10 | 50 | 100 |
| Hexyl 2-methyl butyrate | Female | 0.33 ± 0.16d | 0.78 ± 0.23c | 1.55 ± 0.46b | 2.45 ± 0.54a | 2.53 ± 0.38a |
| | Male | 0.21 ± 0.09c | 0.48 ± 0.13b | 0.95 ± 0.24b | 1.86 ± 0.48a | 2.24 ± 0.53a |
| 2-ethylhexanol | Female | 0.09 ± 0.01e | 0.34 ± 0.02d | 0.68 ± 0.23c | 1.43 ± 0.33b | 2.33 ± 0.46a |
| | Male | 0.08 ± 0.01e | 0.22 ± 0.03d | 0.52 ± 0.12c | 0.95 ± 0.24b | 1.46 ± 0.32a |
| Humulene | Female | 0.11 ± 0.01d | 0.45 ± 0.08c | 0.88 ± 0.12b | 1.56 ± 0.28a | 1.64 ± 0.33a |
| | Male | 0.10 ± 0.01d | 0.29 ± 0.04c | 0.68 ± 0.11b | 0.92 ± 0.21a | 0.98 ± 0.18a |
| Leaf acetate | Female | 0.09 ± 0.01e | 0.33 ± 0.04d | 0.59 ± 0.15c | 0.87 ± 0.21b | 1.59 ± 0.26a |
| | Male | 0.11 ± 0.02d | 0.31 ± 0.03c | 0.38 ± 0.09c | 0.56 ± 0.15b | 0.78 ± 0.16a |
| Aristolochene | Female | 0.15 ± 0.03e | 0.41 ± 0.14d | 0.75 ± 0.20c | 1.18 ± 0.29b | 1.62 ± 0.35a |
| | Male | 0.12 ± 0.02d | 0.33 ± 0.05c | 0.35 ± 0.10c | 0.66 ± 0.12b | 1.11 ± 0.23a |
| Longifolene | Female | 0.23 ± 0.04e | 0.85 ± 0.20d | 1.55 ± 0.35c | 2.05 ± 0.38b | 2.44 ± 0.41a |
| | Male | 0.18 ± 0.02e | 0.44 ± 0.16d | 0.96 ± 0.22c | 1.86 ± 0.28b | 2.18 ± 0.35a |

**Notes.**
Data in the table are means ± SE, different lowercase letters in the same row indicate significant differences ($P < 0.05$).

## Olfactory behavioral responses of *T. vishnou gigantina* to six herbivore-infested *H. rhamnoides* volatiles

The concentration of volatile substances used for measuring olfactory behavioral response was the highest concentration in EAG response, at 100 μg/μL (*Xie et al., 2024*). The test results revealed significant differences in the olfactory behavioral responses of female and male imagoes to the six herbivore-infested volatiles (Table 4). The compounds with the highest behavioral trends were hexyl 2-methyl butyrate, with trend rates of 82.50% and 76.92%, followed by longifolene (80.00% and 68.42%). The volatile with the weakest olfactory behavioral response was leaf acetate (43.24% and 46.15%). The $\chi^2$ test results confirmed that the total number of *T. vishnou gigantina* imagoes attracted to hexyl 2-methyl butyrate and longifolene was significantly different from that attracted to the blank control ($P < 0.05$). This indicates that these two compounds have a chemotactic effect on female and male *T. vishnou gigantina* imagoes. Only female *T. vishnou gigantina* imagoes showed a significant behavioral preference for 2-ethylhexanol, with a convergence rate of 76.31% compared to the blank control. In contrast, three volatiles: humulene, Leaf acetate, and Aristolochene, had no significant attraction effect on *T. vishnou gigantina* imagoes ($P > 0.05$). Overall, the behavioral trends of female *T. vishnou gigantina* imagoes to the six standard compounds in the descending order of attraction: hexyl 2-methyl butyrate, longifolene, 2-ethylhexanol, aristolochene, humulene, and leaf acetate; the trends for male imagoes were hexyl 2-methyl butyrate, longifolene, humulene, leaf acetate, 2-ethylhexanol, and aristolochene.

## DISCUSSION

EAG response and olfactory behavioral response tests can identify factors in plant volatiles that have chemo-ecological effects on insects (*Bhowmik et al., 2016*). The test results

**Table 4  Behavioral responses of *T. vishnou gigantina* to six compounds from herbivore-infested *H. rhamnoides*.**

| Compounds | Sex | Total numbers | | | Luring rate (%) | $\chi^2$ test ($P$) |
|---|---|---|---|---|---|---|
| | | Odor arm | Control arm | No response | | |
| Hexyl 2-methyl butyrate | Female | 33 | 6 | 1 | 82.50 | 4.6213[*] ($P = 0.0256$) |
| | Male | 30 | 7 | 2 | 76.92 | 4.4453[*] ($P = 0.0375$) |
| 2-ethylhexanol | Female | 29 | 7 | 2 | 76.31 | 4.4322[*] ($P = 0.0366$) |
| | Male | 18 | 17 | 5 | 45.00 | 0.1543[NS] ($P = 0.7015$) |
| Humulene | Female | 17 | 18 | 4 | 43.59 | 0.1325[NS] ($P = 0.7256$) |
| | Male | 18 | 15 | 5 | 47.36 | 0.2463[NS] ($P = 0.6954$) |
| Leaf acetate | Female | 16 | 19 | 2 | 43.24 | 0.1312[NS] ($P = 0.7421$) |
| | Male | 18 | 18 | 3 | 46.15 | 0.2135[NS] ($P = 0.6854$) |
| Aristolochene | Female | 16 | 18 | 2 | 44.44 | 0.1468[NS] ($P = 0.6895$) |
| | Male | 15 | 19 | 2 | 41.67 | 0.1025[NS] ($P = 0.7652$) |
| Longifolene | Female | 32 | 7 | 1 | 80.00 | 4.4547[*] ($P = 0.0242$) |
| | Male | 26 | 8 | 4 | 68.42 | 3.8856[*] ($P = 0.0415$) |

**Notes.**

Significance levels of $\chi^2$ test are indicated by NS ($P > 0.05$) and * ($P < 0.05$).

showed that at specific concentrations, six standard compounds can induce EAG and olfactory behavioral responses in male and female *T. vishnou gigantina* imagoes. Therefore, herbivore-infested *H. rhamnoides* volatiles play a positive role in attracting *T. vishnou gigantina* to hosts. Similarly, volatiles induced by *Phauda flamman* larvae have a strong attraction effect on male and female imagoes.

The results showed that after *H. rhamnoides* leaves were damaged by *T. vishnou gigantina*, the relative content of terpenoids and esters in their volatiles increased significantly. These herbivore-infested volatiles had a notable attraction to male and female imagoes. In particular, Hexyl 2-methyl butyrate showed a stronger attraction to female *T. vishnou gigantina* imagoes than males, indicating that *T. vishnou gigantina* has a specific response to herbivore-infested *H. rhamnoides* volatiles. This result is similar to the findings on numerous phytophagous insects such as Hemiptera (*Badra et al., 2021*), Coleoptera (*Ballhorn, Kautz & Heil, 2013*), and Diptera (*Hern & Dorn, 2004*). Therefore, herbivore-infested volatiles are beneficial for controlling insect oviposition and mating.

In the EAG response test, the highest concentration of the six volatiles was 100 µg/µL. Compared with other concentrations in EAG and behavioral tests, Hexyl 2-methyl butyrate showed the strongest attraction (*i.e.,* chemotactic) to female and male imagoes, followed by 2-ethylhexanol and longifolene. 2-ethylhexanol showed a significant attraction to female imagoes, although there was no significant attraction to males. In the *H. rhamnoides* volatiles induced by *T. vishnou gigantina*, the semiochemicals hexyl 2-methyl butyrate and longifolene showed notable stimulus responses. Herbivore-infested volatiles can target and lure adults of different sexes and play an important role in adult host localization. The stimulation responses of these single informative compounds are beneficial to the

control of insect egg-laying and mating, as well as attracting or avoiding insects, which provides a scientific basis for better monitoring and controlling *T. vishnou gigantina* using plant volatiles in the future. However, the olfactory mechanism of phytophagous insects is complex, and the EAG response to a single plant volatile is significantly different from that to a full set of plant volatiles (*Hare, 2011*). Future studies should investigate the physiological and behavioral responses of *T. vishnou gigantina* based on a combination of different herbivore-infested plant volatiles.

The responses of female and male *T. vishnou gigantina* imagoes to the same *H. rhamnoides* volatiles were basically the same. Under the specified concentration gradient, EAG and olfactory behavioral responses significantly increased with the increase in volatile concentration. However, the EAG response to herbivore-infested volatiles at the same concentration showed certain differences between female and male *T. vishnou gigantina* imagoes. For example, in a study on volatiles from *Eucalyptus* leaves, limonene at a dose of 100 $\mu$g had a significant oviposition repellent effect on *Helicoverpa armigera* egg-holding female moths (*Yuan, Huang & Chen, 2021*). This maybe related to the number and functional categories of sensors on the antennae of imagoes, and further research is needed in the future.

To study the behavioural responses of *T. vishnou gigantina* to herbivore-infested plant volatiles, and determine the relationship between herbivore-infested plant volatiles and *T. vishnou gigantina* behaviours, the herbivore-infested key volatiles can be used as attractants to improve the efficacy of insecticides, or the behaviors of herbivore can be interfered in host recognition and host localization, mating and searching for oviposition sites (*Ingrao, Walters & Szendrei, 2019*). This will help develop new methods to control *T. vishnou gigantina*, and provide a theoretical basis for effective controlling other herbivores and developing efficient lure agents.

## CONCLUSION

This study investigated the significant effects of herbivore-infested *H. rhamnoides* volatiles on the electroantennogram and olfactory behavioral of *T. vishnou gigantina*. This information will be of great value for the development of herbivore insect control methods, as they reveal the chemical compounds released by *H. rhamnoides* upon initial infestation, which will help further attract *T. vishnou gigantina* in the future.

## ACKNOWLEDGEMENTS

We are grateful to all students and staff in the Key Laboratory of Ecological Restoration in Shanbei Mining Area and Plant Pest Control in Yulin City, Yulin University, Shaanxi, China for their assistance.

### Funding

This work was supported by the Natural Science Foundation of China (No. 32260398) and Key Research Program of Shaanxi Provincial Department of Education (No. 21JS045). The funders had no role in study design, data collection and analysis, decision to publish, or preparation of the manuscript.

### Grant Disclosures

The following grant information was disclosed by the authors:
Natural Science Foundation of China: 32260398.
Key Research Program of Shaanxi Provincial Department of Education: 21JS045.

### Competing Interests

The authors declare there are no competing interests.

### Author Contributions

- Yonghua Liu conceived and designed the experiments, performed the experiments, authored or reviewed drafts of the article, and approved the final draft.
- Kexu An analyzed the data, prepared figures and/or tables, and approved the final draft.
- Shuo Tang analyzed the data, prepared figures and/or tables, and approved the final draft.
- Jiangshuai Feng conceived and designed the experiments, authored or reviewed drafts of the article, and approved the final draft.
- Xiongfei Yan performed the experiments, authored or reviewed drafts of the article, and approved the final draft.

### Data Availability

   The raw measurements are available in the Supplementary File.

### Supplemental Information

Supplemental information for this article can be found online at http://dx.doi.org/10.7717/peerj.20124#supplemental-information.

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
