# Peer review of "Electroantennogram and Olfactory behavioral responses of *Trabala vishnou gigantina* (Lepidoptera: Lasiocampidae) to herbivore induced *Hippophae rhamnoides* leaf volatiles"

_PeerJ, doi:10.7717/peerj.20124_

## Round 0.1 · original submission · Major Revisions

Dear Dr. Liu, I ask you to carefully respond to the reviewers' fundamental comments.

Reviewer 1 ·

Basic reporting

In the presented study, the researchers identified six key volatile compounds that differed between healthy and infested leaves. The responses of adult T. vishnougigantina moths to these volatiles were evaluated based on electroantennogram (EAG) and olfactory behavioural response experiments.
(line 67 - Collection of H. rhamnoides samples) - It is not known whether the so-called healthy leaves came from the same plant as the leaves affected by feeding.
The absence of signs of feeding on a leaf does not necessarily indicate that it is healthy if it was collected from the same plant as the damaged leaves. The defensive response of a given plant may already have occurred; this factor was not considered in these studies. To correctly interpret the results, detailed studies are needed to determine which H. rhamnoides compounds are primary and which are secondary metabolites. There is no such division in Table 2. The study result is only partial. In addition, other data, such as higher concentrations of compounds in the so-called healthy leaves, were not taken into account.

Some parts of the discussion are linguistically and stylistically incomprehensible (lines 226- 232).

There is also a lack of citation of a very similar publication (Liu et al., 2022). 'Electrophysiological and behavioural responses of the moth Trabala vishnou gigantina (Yang, 1978) (Lepidoptera: Lasiocampidae) to volatiles produced by the plant Hippophae rhamnoides L. (Elaeagnaceae). The Pan-Pacific Entomologist, 98(1): 18–27 (2022).

Experimental design

The research question should be clarified by taking into account publications by other authors with similar titles and aims, such as Liu et al. (2022). 'Electrophysiological and behavioural responses of the moth Trabala vishnou gigantina (Yang, 1978) (Lepidoptera: Lasiocampidae) to volatiles produced by the plant Hippophae rhamnoides L. (Elaeagnaceae). The Pan-Pacific Entomologist, 98(1): 18–27 (2022).

Line 224: T. vishnourhamnoides -what is this?
Line 205: The marked sentence is not understandable.
Line 53-54: That is not true. You should cite the relevant publications and address the question of whether this problem still exists.

Validity of the findings

no comment

Additional comments

no comment

Annotated reviews are not available for download in order to protect the identity of reviewers who chose to remain anonymous.

·

Basic reporting

The authors attempted to study the response of a lepidopteran pest, Trabala vishnou gigantina towards the volaties emanated from damaged leaves of sea buck thorn in their manuscript “Electroantennogram and Olfactory Behavioral Responses of Trabala vishnou gigantina (Lepidoptera: Lasiocampidae) to Six Pest-induced Hippophae rhamnoides volatiles”

Title : The title can be written as “Electroantennogram and Olfactory Behavioral Responses of Trabala vishnou gigantina (Lepidoptera: Lasiocampidae) to herbivore induced Hippophae rhamnoides leaf volatiles”
why the authors have not used the word “herbivore induced volatiles “ at any point in the manuscript- any reason .
Introduction. It is brief and apt.

Experimental design

Materials and Research Method
Line 63: Criteria for distinguishing sexes may please be provided. Reference. please .
Lines 67: How was the leaf sample selected. Was it natural infestation. Was there any other pest infestation.
Line 100: what is section 1.3??
Line 128: Is it 20 per sex per replication . Have you pooled the replications for presenting the results. Details may be mentioned.
Line 130: Please mention how many concentrations were tested for each volatile in the methodology.
If 20 insects were tested were the same insects used for each replication of different set of insects. In these behavioural experiments in some of the treatments were there any missing insects.
data in table 4 - is it chi square test or t test to be followed . please check

The raw data for all experiments was provided .

Validity of the findings

Results and discussion:
Data analysis: analysis for the data presented in table 3 is not clear.
Lines 139-144. Table 2 includes many more volatiles which were significantly different in their % relative content..
For example: 10-Undecenal, 10,10-Dimethyl-2,6-dimethylene bicyclo[7.2.0] undecane, Butanoicacid, butyl ester, cis-3-Hexenyl pyruvate, etc. these compounds were not mentioned at all, Is there any specific reason for not considering these compounds and choosing the studied 6 volatiles. The reasons there of may be mentioned in the manuscript. There is a decrease in the relative content of some compounds after infestation. So in the results section, the point may also be mentioned.
Table 2 does not have significance marked. The points to be addressed are mentioned in the track changes.
Line 2025-206 : the sentence says “ 2-ethylhexanol produced a significant attracting effect on female imagoes, although there was no significant attraction to males”. but data in table 3 shows that attraction was at par
References:
Line 271-273 : Liu, Q.S., Hu, X.Y., Su, S.L., Ning, Y.S., Peng, Y.F., Ye, G.Y., Lou, Y.G., Turlings, T.C.J., Li, Y.H.Cooperative herbivory between two important pests of rice .Nature Communications, 12: 6772, (2021b). DOI:10.1038/s41467-021-27021-0
Please check this citation .

Additional comments

General Comments
The spacing between words may be checked once again.
English language may be checked.
you a look into the statistical analysis and how to present the data.
Detailed foot notes may be provide below the tables .
A reviewed manuscript in track changes is enclosed for favour of information.

·

Basic reporting

Research aimed at identifying and studying chemical molecules that plants naturally produce in response to pest damage provides the basis for the development of modern plant protection methods. The topic of the manuscript under review is relevant. The experiment is correct. The results and conclusions are substantiated and supported by statistical analysis. In general, the article is formatted in accordance with the requirements. Despite this, the manuscript contains some shortcomings and technical comments.

I recommend changing and shortening the title of the manuscript. The title should be short and concise. For example, one of the title options: "Behavioral responses of Trabala vishnou gigantina (Lepidoptera: Lasiocampidae) to volatiles induced by Hippophae rhamnoides."

Experimental design

In the abstract, sentences 2–4 (lines 9–16) should be moved to the Introduction section. The text in the abstract should describe only the research results.

The Introduction section is insufficient in volume and contains few literary citations. I recommend that authors familiarize themselves with the literature on this topic in more detail.

In the abstract and text of the manuscript, at the first mention of an animal organism, in addition to the full Latin name, it is necessary to indicate the author's surname and the year of description of the species in accordance with the International Code of Zoological Nomenclature (lines 8, 34).

Throughout the text, the Latin names of species are merged and must be separated. When citing, a comma must be placed after the authors' surnames before the year of publication.

The materials and methods are described in great detail. They take up a disproportionately large volume of the manuscript. It needs to be cut down.

Validity of the findings

It is difficult to judge the reliability of the results, since the replication and sample are described superficially (lines 106–109). How many individuals participated in the experiments? Six females and six males? This is an insufficient sample to establish statistically reliable conclusions.

The sentence (lines 188–189) requires support with a literary citation.

I recommend that the authors rephrase the sentence (lines 225–230) and break it into parts. Remove the repeated phrases in the sentence.

Describe the limitations of the study and add it to the "Discussion" section.

The conclusions need to be expanded a little. The sentence (lines 234–235) needs to be rephrase. Electroantennogram is a research method, it cannot be affected by volatile substances.

Additional comments

No comments.

---

## Round 0.2 · Minor Revisions

Dear Dr. Liu, I ask you to make minor corrections to the manuscript. This will allow it to be accepted for publication as soon as possible.

Reviewer 1 ·

Basic reporting

Dear authors,
In the manuscript, the species' name needs careful correction.
The name of the genus and species must be written separately.
There is also a name mistake in Table 4.
The corrections are marked in the text.
Apart from that, I have no other comments.

Experimental design

no comment

Validity of the findings

no comment

Additional comments

no comment

Annotated reviews are not available for download in order to protect the identity of reviewers who chose to remain anonymous.

Reviewer 4 ·

Basic reporting

1. Line 21-23: Correct the contradictory statements about Hexyl 2-methyl butyrate's attraction to males (initially claimed strongest but later stated insignificant) and remove the repeated phrase 'followed by Hexyl 2-methyl butyrate' to ensure logical consistency.

Experimental design

2.Clarify whether tested adults were mated/unmated and age-controlled (beyond "1–3 days old"), as this affects olfactory sensitivity.
3.The experimental group used liquid paraffin (Section 2.3), while the control group used n-hexane (Section 2.5). The difference in solvents might interfere with the behavioral results. The same solvent (such as liquid paraffin) should be used for the control.

Validity of the findings

4.The paper did not report the normality test of the EAG data, which will affect the reliability of the Duncan multiple comparison results. It is recommended that the authors conduct additional analysis or clearly explain the testing process.
5.Line 145-146: Revise Section 3.2 to resolve the contradictory concentration descriptions (1 µg/µL vs. volatile concentration 0.1 µg/µL).

Additional comments

Emphasize how this work advances beyond Liu et al. 2022 (cited) by focusing on herbivore-induced volatiles.

---

## Round 0.3 · Minor Revisions

Dear Dr. Liu, I ask you to make minor changes in the note under table 2. The standard for world science is three levels of significance of differences: 0.05 denotes *, 0.01 denotes ** and 0.001 denotes ***. I ask you to apply these three thresholds and recalculate the significance of differences between the samples of this table. All figures in this table should be rounded to hundredths. This will also lead to minor changes in the text of the article. I hope that there will be no errors in the article and it will be recommended for publication as soon as possible.

Reviewer 1 ·

Basic reporting

no comment

Experimental design

no comment

Validity of the findings

no comment

Additional comments

The manuscript has been improved according to all suggestions. I have no further comments.

·

Basic reporting

The authors took into account all the comments I made earlier. They made changes to the text. I believe that this article can be recommended for publication.

Experimental design

The authors took into account all the comments I made earlier. They made changes to the text. I believe that this article can be recommended for publication.

Validity of the findings

The authors took into account all the comments I made earlier. They made changes to the text. I believe that this article can be recommended for publication.

Additional comments

No comments.

Reviewer 4 ·

Basic reporting

There are no more issues. It can be published now.

Experimental design

No comment.

Validity of the findings

No comment.

Additional comments

No comment.

---

## Round 0.4 · accepted · Accept

Dear Dr. Liu, I congratulate you on the acceptance of this article for publication.